# PROGRESSIVE PROMPTS: CONTINUAL LEARNING FOR LANGUAGE MODELS

**Anastasia Razdaibiedina**[*][†]**, Yuning Mao**[‡]**, Rui Hou**[‡]**,**

**Madian Khabsa**[‡]**, Mike Lewis**[‡]**, Amjad Almahairi**[‡]

[†]University of Toronto & Vector Institute
[‡]Meta AI
```
anastasia.razdaibiedina@mail.utoronto.ca,
{yuningm,rayhou,mkhabsa,mikelewis,aalmah}@meta.com
```

## ABSTRACT

We introduce Progressive Prompts – a simple and efficient approach for continual learning in language models. Our method allows forward transfer and resists catastrophic forgetting, without relying on data replay or a large number of task-specific parameters. Progressive Prompts learns a new soft prompt for each task and sequentially concatenates it with the previously learned prompts, while keeping the base model frozen. Experiments on standard continual learning benchmarks show that our approach outperforms state-of-the-art methods, with an improvement >20% in average test accuracy over the previous best-preforming method on T5 model. We also explore a more challenging continual learning setup with longer sequences of tasks and show that Progressive Prompts significantly outperforms prior methods.

## 1 INTRODUCTION

Learning a long sequence of tasks while gaining experience and avoiding forgetting remains a key feature of human-level intelligence. Although pretrained language models have largely succeeded in learning on a single task, their performance degrades in scenarios where multiple tasks are encountered sequentially, also known as *continual learning* (CL) (de Masson D'Autume et al., 2019; Huang et al., 2021). Two major challenges arise in CL: (1) avoiding *catastrophic forgetting*, i.e., loss of the knowledge acquired from previous tasks after learning new ones (McCloskey & Cohen, 1989; Ratcliff, 1990), and (2) allowing *forward transfer*, i.e., leveraging the knowledge from past tasks for efficient learning of new tasks.

Typical CL approaches for language models train a model on all tasks, which ensures forward transfer but also leads to forgetting. These methods use data replay or add regularization constraints (Huang et al., 2021; de Masson D'Autume et al., 2019; Sun et al., 2019), but they still suffer from forgetting due to inevitable changes in parameters shared between tasks. Other approaches, such as progressive networks (Rusu et al., 2016), can alleviate catastrophic forgetting completely while supporting forward transfer, but are computationally expensive because they add a new copy of the model for each task. This can be especially intractable for large-scale language models with billions of parameters, which have become a standard in the NLP field (Zhang et al., 2022).

In this paper, we introduce **Progressive Prompts** – a novel CL approach for language models that supports forward transfer without forgetting. Our method is inspired by progressive networks, but is significantly more memory-efficient because it only learns a fixed number of tokens, or *prompt*, for each new task. Learning a prompt to adapt language models on a single downstream task was introduced in *prompt tuning* (Lester et al., 2021), and was shown to match the performance of full model finetuning while training <0.01% of the parameters. In Progressive Prompts, we learn a separate prompt for each incoming task and sequentially concatenate it with previously learned prompts. Importantly, we share input tokens across all tasks and progressively prepend new prompts while keeping previous prompts frozen (see Figure 1). Our method can: 1) alleviate catastrophic

---

[*]Work done during Meta AI research internship.

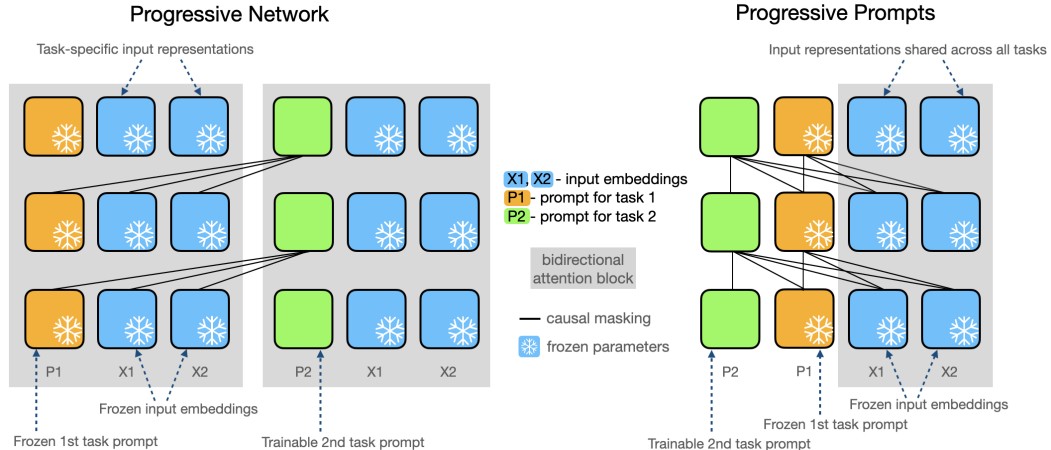

Figure 1: Illustrating our proposed method *Progressive Prompts* and contrasting it with a simple adaptation of progressive networks using prompt tuning. In the simple adaptation of progressive networks we learn a separate prompt and repeat the frozen input embeddings for each new task. This setup requires repeating input tokens for each task. In Progressive Prompts we use the same input and progressively append new prompt for each new task. Prior task prompts are not modified by the addition of new prompts.

forgetting by preserving the knowledge acquired in previous prompts, and 2) transfer knowledge to future tasks by sequentially learning new prompts given previous ones. We also introduce a new technique for prompt embedding reparameterization (Li & Liang, 2021). We show that by passing the prompt embeddings through a residual MLP we can stabilize prompt tuning and improve its performance.

We run extensive experiments on standard CL benchmarks for text classification, and show that Progressive Prompts outperforms state-of-the-art approaches on both BERT and T5 architectures (Devlin et al., 2018; Raffel et al., 2020). We show over 20% improvement over the current SOTA for T5 model (Qin & Joty, 2021). Furthermore, we run experiments on a more challenging CL setup with longer task sequences, and show that our method outperforms prior approaches for T5 and BERT architectures.

Our main contributions in this paper are as follows:

- We propose a novel CL approach, *Progressive Prompts*, that alleviates catastrophic forgetting and supports knowledge transfer to future tasks – all while learning $< 0.1\%$ of the total parameters.
- Progressive Prompts is suitable for any transformer-based architecture. We show that it significantly outperforms prior SOTA methods on standard CL benchmarks for both BERT and T5 models.
- We propose a more challenging CL setup encompassing 15 text classification tasks and show that our method significantly outperforms prior methods.

## 2 BACKGROUND

### 2.1 FINETUNING

The predominant technique for adapting the language model to a downstream task $T$ is *finetuning*, when all parameters $\Theta$ of the model (initialized from some pre-trained weights) are updated during task adaptation (Devlin et al., 2018; Zhang et al., 2020).

Consider a classification task $T$ with input text $x$, and output scalar label $y$. Here $p_\Theta$ is a probability distribution of output classes parameterized by the weights, $\Theta$, of the language model. In finetuning,

we perform gradient updates according to the following log-likelihood objective:

$$\max_{\Theta} \sum_{x,y \in T} \log p_{\Theta}(y|x)$$

Despite its simplicity and effectiveness, finetuning updates all model parameters, and when there are multiple downstream tasks it requires storing a separate finetuned model for each of them.

## 2.2 PROMPT TUNING

Recent works explored *prompt tuning* (Lester et al., 2021) as a lightweight alternative for finetuning, when instead of training the whole model, a series of virtual tokens – or *soft prompt* – is trained. In prompt tuning, a soft prompt $P$ is prepended to the input text $x$, while keeping other parameters frozen. In this case, the model parameters $\Theta$ are composed of two sets: 1) the pretrained language model parameters $\theta$ which are *fixed* on downstream tuning, and 2) prompt parameters $\theta_p$ which are fine-tuned on the given task. Since prompt $P$ has its own dedicated parameters $\theta_p$, we are not restricted by the original model parametrization $\theta$, and can perform gradient updates on the prompt parameters independently.

Given a task $T$ and a probability distribution $p_{\theta,\theta_p}$ of output classes parameterized by $\theta$ and $\theta_p$, the learning objective here is to maximize the log-likelihood of $y$ given the input text $x$ concatenated with the soft prompt $P$, or

$$\max_{\theta_P} \sum_{x,y \in T} \log p_{\theta,\theta_P}(y|[P;x])$$

## 2.3 CONTINUAL LEARNING

In this work, we focus on a continual learning setup, where the language model is presented with a sequence of $m$ text classification tasks $(T_1, ..., T_m)$. In each task we have a set if i.i.d. training examples $(x_i, y_i)_{i=1}^N$, where $x_i$ is an input text and $y_i$ is a label from a set of predefined labels $Y_k$ associated with the task $T_k$. We assume that the model is parameterized by $\Theta$ and has access to the task identity in both training and inference. Hence, the learning objective across all tasks becomes:

$$\max_{\Theta} \sum_{k=1}^{m} \sum_{x,y \in T_k} \log p_{\Theta}(y|x)$$

The most straightforward approach for continual learning is finetuning, which sequentially optimizes loss for task $k$, $k \in 1..m$ by updating all model parameters: $\mathcal{L}_k(\Theta) = -\sum_{x,y \in T_k} \log p(y|x, \Theta)$. Continual finetuning will support forward knowledge to future tasks, but will result in catastrophic forgetting, where performance on the earlier tasks decreases after learning new tasks, and eventually leads to a higher generalization loss (Kirkpatrick et al., 2017; de Masson D'Autume et al., 2019; McCloskey & Cohen, 1989).

## 3 METHOD

**Progressive Prompts** We propose *Progressive Prompts*, a continual learning approach which progressively learns a prompt $P_k$ for each new task $T_k$ (Figure 1). With Progressive Prompts we learn a separate prompt $P_k$ for task $T_k$, and concatenate it with all previously learned prompts $P_i, i < k$, before prepending to the input embeddings. During training, parameters $\theta$ of the language model are always frozen, and parameters $\theta_{P_k}$ corresponding to prompt $P_k$ are only trainable during learning $T_k$, and frozen afterwards.

The training objective for task $T_k$ ($k \in \{1...m\}$) is to find prompt parameters $\theta_{P_k}$ that minimize the negative log probability of training examples under our progressive prompt and the frozen base model:

$$\mathcal{L}(\theta_{P_k}) = -\sum_{x,y \in T_k} \log p(y|[P_k, ..., P_1, x], \theta, \theta_{P_1}, ..., \theta_{P_k})$$

Such a progressive setup allows to achieve two goals for efficient CL: (1) it successfully eliminates *catastrophic forgetting*, and (2) allows *forward transfer* for subsequent tasks. Since Progressive

Prompts trains a separate prompt for each encountered task without modifying its parameters when new tasks are learned, old tasks do not suffer from forgetting. In addition to that, prompts learned on previous tasks allow information re-use for future tasks. A similar phenomenon has been shown by Vu et al. (2021) – prompts learned on informative source tasks served as a good initialization for other downstream tasks.

**Embedding reparameterization** Li & Liang (2021) have shown that directly optimizing prompt parameters can lead to training instability, while reparameterizing prompt embeddings matrix through a multi-layer perceptron (MLP) can improve performance. At the same time, Liu et al. (2021) report that prompt embedding reparameterization can hinder performance on certain tasks and is highly sensitive to its initialization. To solve this issue, we propose adding a residual connection to the MLP reparameterization. This residual connection improves optimization, because it avoids learning an identity mapping by the MLP. More specifically, we reparameterize prompt $P_k$ for task $T_k$ as follows:

$$P_k^{'} = \text{MLP}(P_k) + P_k$$

After training on task $k$ is complete, the reparameterization parameters MLP can be discarded, and prompt embeddings $P_k$ can be replaced by their corresponding projection $P_k^{'}$.

## 4 Experimental Setup

### 4.1 Datasets

**Continual Learning Benchmark** We first evaluate our approach on the widely adopted CL benchmark for language models, which includes five text classification datasets by Zhang et al. (2015): AG News (4 classes, news classification), Amazon reviews (5 classes, sentiment analysis), Yelp reviews (5 classes, sentiment analysis), DBpedia (14 classes, Wikipedia text classification) and Yahoo Answers (10 classes, Q&A classification). We provide the task details in Appendix A.1 and sequences of tasks used in our experiments in Appendix A.2.

Following previous works on CL for BERT model, including IDBR (Huang et al., 2021) and MBPA++ (de Masson D'Autume et al., 2019), for BERT-based experiments we use four different orders of these five tasks. We use the same train and test sets as IDBR (Huang et al., 2021) and MbPA++ (de Masson D'Autume et al., 2019), consisting of 115,000 training and 7,600 test examples for each task. Following Huang et al. (2021), for every task we randomly hold out 500 samples per class from the training set for validation, and use early stopping according to the validation accuracy on all seen tasks.

Following CL setup for T5 model as in LFPT5 (Qin & Joty, 2021), we use three different orders of the AG News, Amazon, Yahoo and DBpedia datasets. We replicate Qin & Joty (2021) few-shot CL setting and sample 16 examples per task for the training set and keep the test sets unchanged.

**Large number of tasks** While previous CL approaches mostly focused on relatively short task sequences of 3-5 tasks (Huang et al., 2021; de Masson D'Autume et al., 2019; Qin & Joty, 2021), a more realistic CL scenario would be long sequences encompassing more tasks. Hence, we create a benchmark of 15 text classification tasks to evaluate the performance of Progressive Prompts along with widely adopted CL approaches. Our benchmark consist of the aforementioned five datasets from CL benchmark, combined with four tasks from GLUE benchmark (MNLI, QQP, RTE, SST2) (Wang et al., 2018), five tasks from SuperGLUE benchmark (Wang et al., 2019) (WiC, CB, COPA, MultiRC, BoolQ), and IMDB movie reviews dataset (Maas et al., 2011). To evaluate performance under different dataset sizes, we create three different versions of each dataset, with 20, 200 and 1000 training samples sampled per class, and report test set performance for each of these settings. Again, we follow Huang et al. (2021) practice and for every task we randomly hold out 500 samples per class from the training set for validation, and use early stopping according to the validation accuracy on all seen tasks. We provide the task details in Appendix A.1 and sequences of 15 tasks used in our experiments in Appendix A.2.

**Transfer learning experiments** We also run ablation experiments to understand the effect of forward transfer with Progressive Prompts. Firstly, we perform experiments on six pairs of transfer learning tasks from similar domains (IMDb & SST2, Amazon & Yelp, QQP & MRPC) following Hendrycks et al. (2020). Secondly, we perform experiments on SuperGLUE benchmark (Wang et al.,

2019) with Progressive Prompts and compare the performance against original per-task prompt tuning by Lester et al. (2021). Following Lester et al. (2021) setup, we use full training sets of eight SuperGLUE tasks and report best validation performance.

## 4.2 BASELINES

We consider nine baseline methods for comparison with ProgressivePrompts:

- **Finetune** (Wang et al., 2020; de Masson D'Autume et al., 2019; Huang et al., 2021): train all model parameters on a sequence of tasks (without adding any regularization constraints or replaying samples from the previous tasks).

- **EWC** (Kirkpatrick et al., 2017): finetune the whole model with a regularization loss that prevents updating parameters that could interfere with previously learned tasks.

- **A-GEM** (Chaudhry et al., 2018): save examples from the past tasks and restrict the gradients used to update the model on new tasks based on the retrieved examples.

- **Experience replay**: finetune the whole model with a memory buffer, and replay samples from old tasks when learning new tasks to avoid forgetting.

- **MBPA++** (de Masson D'Autume et al., 2019): augment BERT with an episodic memory that saves all seen examples. Perform replay during training, and local adaptation during test time.

- **IDBR** (Huang et al., 2021): BERT-specific approach which continuously trains the whole model using data replay and a regularization loss, which applies sentence representation disentanglement into task-specific and task-generic spaces. Current SOTA on CL benchmark with BERT.

- **Per-task prompts** (Lester et al., 2021): train a separate soft prompt for each task, while keeping the original model frozen. This setup will eliminate catastrophic forgetting, since per-task parameters do not change when new tasks are learned, but will not result in forward transfer.

- **PromptTuning** (Lester et al., 2021; Qin & Joty, 2021): train a shared soft prompt sequentially on all tasks, while keeping the original model parameters frozen.

- **LFPT5** (Qin & Joty, 2021): continuously train a soft prompt that simultaneously learns to solve the tasks and generate training samples, which are subsequently used in experience replay. Current SOTA on CL benchmark with T5.

## 4.3 IMPLEMENTATION DETAILS

Progressive Prompts is a model-agnostic CL method that can be used with any transformer-based model. In our experiments, we use two language models adopted by the previous lines of works in CL for NLP: encoder-only **BERT** model (Devlin et al., 2018) and encoder-decoder **T5** model (Raffel et al., 2020). To compare Progressive Prompts to the recent CL approaches implemented specifically with BERT, we use the pre-trained BERT-base model as in IDBR and MBPA++ methods (Huang et al., 2021; de Masson D'Autume et al., 2019). For comparison with LFPT5 approach (Qin & Joty, 2021) we use pre-trained T5-large[1] model.

**BERT** Following Devlin et al. (2018), to predict the class of input text $x$, we use the representation of its first token $h_{[CLS]}$ (which is encoded by a special beginning-of-a-sentence symbol, [CLS]) as a sentence representation. Here $h$ is the whole-input representation matrix from BERT encoder. We apply a linear transformation parametrized by $w$ and a softmax function to obtain the classification probabilities over classes $c \in \{1...\mathcal{C}\}$:

$$p(y = c|h) = \frac{\exp(w_c h_{[CLS]})}{\sum_{y \in \mathcal{C}} \exp(w_y h_{[CLS]})}$$

In BERT experiments we train a separate linear head in addition to prompt embeddings for each task, using cross-entropy loss between the predicted classification probabilities and ground truth class labels.

---

[1]Although the original prompt tuning approach reports better performance with T5 v1.1 compared to T5, several subsequent works find version v1.1. less stable for prompt tuning compared to the original T5 and report worse performance (Karimi Mahabadi et al., 2021). Therefore, in this work we use the original T5 model.

**T5** Following Raffel et al. (2020) and Lester et al. (2021), we adopt text-to-text formulation for all T5 experiments, where classification labels are mapped into words (e.g. 0/1 could be encoded as "True"/"False"). T5 model applies a multi-headed self-attention over the input tokens followed by position-wise feed-forward layers to produce an output distribution over target tokens. We train prompt embeddings for T5 model using cross-entropy loss. We report the rest of the implementation details in Appendix A.3.

**Prompt length** For all Progressive Prompt experiments on BERT, we set single-task prompt length to 20 tokens and apply prompt reparameterization with a two-layer residual MLP. For T5 experiments, we use single-task prompt length to 10 tokens in case of long-sequence experiments and 50 in case of T5 experiments (so that total Progressive Prompt length more closely matches LFPT5 prompt length of 300). We report more experimental details in Appendix A.4.

## 5 EXPERIMENTAL RESULTS

For all CL experiments we evaluate methods after training on all tasks and report averaged test set scores across all tasks. Metrics used for different tasks are shown in Appendix A.1. We report Progressive Prompts performance with BERT-base and T5-large models, and compare it to the existing CL approaches used with these models.

| Method | DR | Order 1 | 2 | 3 | avg |
|---|---|---|---|---|---|
| Finetune$^\diamond$ | | 18.9 | 24.9 | 41.7 | 28.5 |
| Replay | ✓ | 35.4 | 37.1 | 41.5 | 38.0 |
| EWC$^\diamond$ | | 39.0 | 38.0 | 44.8 | 40.6 |
| LFPT5*$^\diamond$ | ✓ | 47.6 | 52.6 | 57.9 | 52.7 |
| ProgPrompt* | | **75.2** | **75.0** | **75.1** | **75.1** |
| Per-task Finetune | | 70.0 | 70.0 | 70.0 | 70.0 |

(a) Results with T5-large.

| Method | DR | Order 4 | 5 | 6 | 7 | avg |
|---|---|---|---|---|---|---|
| Finetune$^\dagger$ | | 14.8 | 27.8 | 26.7 | 4.5 | 18.4 |
| Replay$^\dagger$ | ✓ | 67.2 | 64.7 | 64.7 | 44.6 | 57.8 |
| A-GEM$^\dagger$ | ✓ | 70.6 | 65.9 | 67.5 | 63.6 | 66.9 |
| MBPA++$^\dagger$ | ✓ | 70.8 | 70.9 | 70.2 | 70.7 | 70.6 |
| IDBR$^\ddagger$ | ✓ | 75.9 | 76.2 | 76.4 | 76.7 | 76.3 |
| ProgPrompt* | | **78.0** | **77.7** | **77.9** | **77.9** | **77.9** |
| Per-task Finetune | | 73.9 | 73.9 | 73.9 | 73.9 | 73.9 |

(b) Results with BERT-base.

Table 1: Summary of the results on two standard CL benchmarks with T5 and BERT models. Averaged accuracy after training on the last task is reported. All results are averaged over 3 runs. For T5 experiments we followed Qin & Joty (2021) protocol and used few-shot CL setting. Methods marked with * only train a soft prompt while keeping the model frozen, other methods train the entire model. DR denotes whether the method requires data replay. $^\diamond$, $^\dagger$ and $^\ddagger$ denote results from Qin & Joty (2021), de Masson D'Autume et al. (2019) and Huang et al. (2021) respectively.

### 5.1 RESULTS ON STANDARD CONTINUAL LEARNING BENCHMARKS

**T5 benchmark**. Table 1a compares Progressive Prompts performance on the few-shot CL benchmark for T5 model with the existing CL approaches, including previous SOTA – LFPT5 (Qin & Joty, 2021). Notably, Progressive Prompts achieve over 20% improvement compared to LFPT5. This drastic increase in accuracy is explained by the fact that Progressive Prompts method does not experience forgetting and allows forward transfer, which is especially useful in few-shot regime. Of note, Qin & Joty (2021) report better performance of LFPT5 in continual learning setting than Adapter-Fusion (Pfeiffer et al., 2020) – a strong parameter-efficient CL approach based on adapters. Hence, we did not include adapter baselines in this study, but instead focused on a more parameter-efficient and better-performing method – LFPT5.

**BERT benchmark**. Table 1b compares performance of Progressive Prompts across four different task orders with existing CL approaches for BERT model, including previous SOTA – IDBR (Huang et al., 2021). We want to note that CL benchmark for BERT focuses on full dataset experiments, contrary to few-shot CL benchmark for T5. Since Progressive Prompts are most efficient in few-shot setting, we observe a larger performance gap with LFPT5 compared to IDBR. Overall, Progressive Prompts further improve performance compared to IDBR, reaching 77.9 average score across all or-

ders. We also note that Progressive Prompts is a model-agnostic approach, contrary to IDBR, which is designed specifically for BERT. Additionally, our method does not require storing data from the previous tasks for future replay, in contrast to the previous commonly used CL approaches for BERT – IDBR and MBPA++. In all of our experiments we use the embedding reparameterization defined in Section 3. We find that this reparameterization helps in stabilizing and accelerating training. Due space limitations, we explore the effect of residual reparameterization in Appendix B.3).

## 5.2 PERFORMANCE WITH LARGE NUMBER OF TASKS

Table 2 compares common CL approaches, including SOTA methods for T5 and BERT models (Qin & Joty, 2021; Huang et al., 2021), on a sequence continual learning with 15 tasks. We report averaged results across three task orders (8, 9 and 10) obtained with T5-Large and BERT-base models. We provide the full non-averaged results for each order in Appendix B.1. To investigate the effect of limited data settings, we perform training on different dataset sizes with 20, 200 and 1000 samples per class.

| Method ↓ | | avg | | Method ↓ | | avg | |
|---|---|---|---|---|---|---|---|
| Num. samples → | 20 | 200 | 1000 | Num. samples → | 20 | 200 | 1000 |
| **T5-Large results** | | | | **BERT-base results** | | | |
| Finetune | 9.7 | 8.3 | 7.4 | Finetune | 31.3 | 42.4 | 41.7 |
| Prompt tuning* | 17.4 | 13.9 | 10.9 | Prompt tuning* | 47.6 | 57.2 | 59.5 |
| Replay | 43.6 | 44.2 | 54.4 | Replay | 36.4 | 47.0 | 49.6 |
| Per-task Prompts* | 69.8 | 75.2 | 77.0 | Per-task Prompts* | 50.6 | 62.4 | 67.2 |
| LFPT5* | 54.3 | 58.2 | 69.2 | IDBR | 36.8 | 47.9 | 52.2 |
| ProgPrompt* | **76.2** | **78.7** | **79.5** | ProgPrompt* | **53.5** | **66.9** | **69.3** |
| MTL | 70.7 | 72.5 | 76.3 | MTL | 56.9 | 67.7 | 69.9 |
| Per-task Finetune | 68.2 | 73.7 | 78.1 | Per-task Finetune | 53.2 | 58.4 | 63.1 |

Table 2: Progressive Prompts outperforms existing continual learning methods on T5-Large and BERT-base models over long sequences of tasks (15 tasks in total). Average results across three different task orders with 20, 200 and 1000 samples per class are shown. MTL denotes multi-task learning. Methods marked with * only train a soft prompt while keeping the model frozen, other methods train the entire model.

Our approach, Progressive Prompts, consistently outperforms all other methods across different data limits, with the strongest improvement in the few-shot setting of 20 samples per class: $+21.9\%$ and $+33.3\%$ over previous SOTA approaches on T5 and BERT models respectively. We also provide more fine-grained performance metrics defined by Lopez-Paz & Ranzato (2017), including forward transfer (FWT) and backward transfer (BWT) scores, as well as evolution of accuracy with learning new tasks in Appendix B.1.

**Attention between prompts** To investigate which prompts from the progressive pool are helpful for learning new tasks, we visualized between-prompt attentions (Figure 2). We used Progressive Prompts trained on order 8 with T5 model. For each of the 15 consecutive tasks, we passed all test examples through T5 encoder, and extracted attention matrices from its last layer (24th layer). We averaged attention scores across all prompt tokens and all heads, and shown the results in Figure 2. Interestingly,

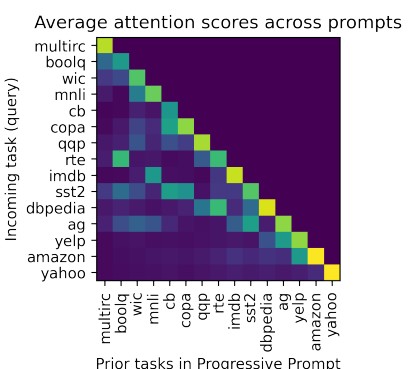

Figure 2: Average attention scores between prompts in Progressive Prompts.

prompts learn to ignore irrelevant tasks and attend to the most similar or informative tasks. For example, prompt learned on Amazon reviews task heavily attends to Yelp reviews prompt, and SST2 prompt attends to the prompt learned on IMDb - another sentiment analysis task.

### 5.3 FORWARD TRANSFER EXPERIMENTS

In this section we study in more detail the *forward transfer* phenomenon achieved with Progressive Prompts. In particular, can a subsequent task prompt exploit the knowledge from a previously learned prompt in learning a new related task?

**Transfer Learning on a pair of tasks** We compare the performance of a single prompt of length $2M$ versus a Progressive Prompt consisting of two $M$-length prompts, $M = 50$ (see Figure 3). We follow a similar transfer learning experimental protocol by Hendrycks et al. (2020) and use pairs of six tasks from similar domains: IMDb and SST2 (2-class sentiment classification), Amazon and Yelp reviews (5-class sentiment classification), MRPC and QQP (paraphrase detection). We use T5-Large model, and study both settings: full-dataset and few-shot (2, 5 and 20 samples per class). Our results are summarized in Table 3 and Figure 4. Overall, we observe that Progressive Prompts achieve forward transfer and outperform the standard prompt tuning across all dataset sizes. The strongest improvements happen in few-shot setting – $+20.4\%$ and $+12.9\%$ relative improvement under 5-shot and 2-shot setup.

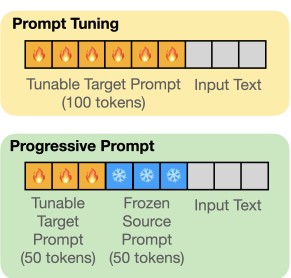

Figure 3: Transfer learning experimental setup. Prompt Tuning: a single prompt of 100 tokens is trained on target task. Progressive Prompt: two prompts of 50 tokens are trained sequentially on source and target tasks.

| N-shot | Task Source → Target | Prompt Tuning (w/o transfer) | Progressive Prompt (w/ transfer) | Relative Improv. |
|---|---|---|---|---|
| All | imdb → sst2 | $93.6_{\pm 2.5}$ | $\mathbf{95.8_{\pm 0.4}}$ | +2.4% |
| | sst2 → imdb | $93.1_{\pm 0.4}$ | $\mathbf{93.3_{\pm 0.3}}$ | +0.2% |
| | amazon → yelp | $61.2_{\pm 0.5}$ | $\mathbf{63.5_{\pm 0.5}}$ | +3.8% |
| | yelp → amazon | $58.2_{\pm 0.9}$ | $\mathbf{59.0_{\pm 0.8}}$ | +1.4% |
| | mrpc → qqp | $\mathbf{91.1_{\pm 0.7}}$ | $90.9_{\pm 0.4}$ | -0.2% |
| | qqp → mrpc | $84.6_{\pm 0.2}$ | $\mathbf{85.0_{\pm 0.8}}$ | +0.5% |
| | Average | 80.3 | $\mathbf{81.2}$ | +1.3% |
| 20/class | imdb → sst2 | $85.0_{\pm 6.0}$ | $\mathbf{91.4_{\pm 1.2}}$ | +7.5% |
| | sst2 → imdb | $\mathbf{93.4_{\pm 0.3}}$ | $93.2_{\pm 0.6}$ | -0.2% |
| | amazon → yelp | $50.2_{\pm 5.0}$ | $\mathbf{52.5_{\pm 2.1}}$ | +4.6% |
| | yelp → amazon | $37.8_{\pm 5.1}$ | $\mathbf{54.9_{\pm 2.1}}$ | +45.2% |
| | mrpc → qqp | $88.6_{\pm 1.3}$ | $\mathbf{89.4_{\pm 1.6}}$ | +0.9% |
| | qqp → mrpc | $84.8_{\pm 0.0}$ | $\mathbf{85.2_{\pm 0.4}}$ | +0.5% |
| | Average | 73.3 | $\mathbf{77.8}$ | +9.8% |
| 5/class | imdb → sst2 | $58.2_{\pm 2.7}$ | $\mathbf{84.4_{\pm 8.1}}$ | +45.0% |
| | sst2 → imdb | $61.1_{\pm 5.0}$ | $\mathbf{91.2_{\pm 2.5}}$ | +49.3% |
| | amazon → yelp | $26.2_{\pm 3.8}$ | $\mathbf{27.3_{\pm 1.4}}$ | +4.2% |
| | yelp → amazon | $24.2_{\pm 2.1}$ | $\mathbf{29.7_{\pm 3.7}}$ | +22.7% |
| | mrpc → qqp | $88.4_{\pm 2.5}$ | $\mathbf{92.0_{\pm 0.3}}$ | +4.1% |
| | qqp → mrpc | $\mathbf{86.9_{\pm 0.6}}$ | $84.3_{\pm 0.6}$ | -3.0% |
| | Average | 57.5 | $\mathbf{68.2}$ | +20.4% |

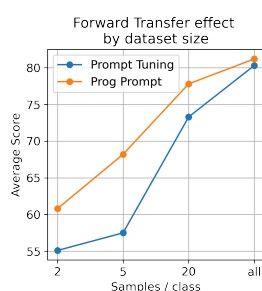

Table 3: Transfer learning results on pairs of six tasks from similar domains (IMDb & SST2, Amazon & Yelp, MRPC & QQP). All results are averaged over 3 runs. We report average score between F1 and accuracy for MRPC and QQP, and accuracy for all other tasks.

Figure 4: Progressive Prompts achieve forward transfer and outperform Prompt Tuning under different dataset sizes. Average scores across six target tasks are reported.

**GLUE → SuperGLUE transfer** We perform additional experiments on SuperGLUE benchmark to compare the forward transfer effect of Progressive Prompts with standard prompt tuning. In the original prompt tuning paper, a separate prompt $P_T$ is trained for each SuperGLUE task $T$ (Lester et al., 2021). $P_T$ is a trainable matrix of $100 \times e$ dimension, where 100 is the prompt length and $e$ if the embedding dimension. $P_T$ is concatenated with the input text $x$ and then fed into the model as $[P_T; x]$. For Progressive Prompt we perform training in two steps. First, we learn a progressive prompt on GLUE benchmark $P_{\text{GLUE}}$. $P_{\text{GLUE}}$ is composed of 6 per-task GLUE prompts with a length of 10 on random task order, and the final dimension of $P_{\text{GLUE}}$ is $60 \times e$. After that, we learn a length-40 prompt $[P_{T_{\text{Prog}}}$ for every SuperGLUE task $T$ and concatenate it with the GLUE

prompt before appending to the input $x$: $[P_{T_{\text{Prog}}}; P_{\text{GLUE}}; x]$ (illustration of the experimental setup is in Appendix B.2).

Following Lester et al. (2021) we train a separate soft prompt for each SuperGLUE task on its full train set with a fixed number of epochs, and report the validation set performance. Similarly to Lester et al. (2021), for each SuperGLUE task use metrics recommended by Wang et al. (2019) and for tasks with two metrics we compute an average of the two. Our results are shown in Table 4 – Progressive Prompt improves the standard prompt tuning performance by 2.7 points on SuperGLUE benchmark. This confirms that progressive prompt concatenation allows knowledge reuse from the previous prompts that were learned on GLUE benchmark.

| Method | SuperGLUE avg. |
|---|---|
| Per-task PT | 74.5±2.2 |
| ProgPrompt | **77.2±1.9** |
| Per-task FT | 81.3±0.6 |

Table 4: Performance of the Prompt Tuning approach vs Progressive Prompts on Super-GLUE. Averaged validation set score across all tasks is shown.

## 6 RELATED WORK

**Continual Learning** Existing continual learning approaches can be broadly organized into three main categories: (i) *replay-based*, (ii) *regularization-based*, and (iii) *architecture-based* (de Masson D'Autume et al., 2019; Huang et al., 2021). Replay-based methods store a subset of data from previous tasks for future rehearsal via experience replay, representation consolidation or constrained optimization. The data can be either stored directly or synthesized by generative models. *Regularization-based* approaches restrict changes of model's parameters to avoid inference with previously learned tasks (Li & Hoiem, 2017; Kirkpatrick et al., 2017). *Architecture-based* approaches learn different set of parameters dedicated for a separate task. Recently, replay-based and regularization-based approaches have been successfully applied for continual learning in language models. While replay-based approaches have shown strong results for classification, question answering and relation extraction tasks, they pose significant memory requirements due to storing large number of samples for rehearsal (Rolnick et al., 2019; Huang et al., 2021). Additionally, for many applications such storage would not be feasible due to privacy settings, when access to past data is not available. Regularization-based approaches are more memory-efficient than replay-based approaches, but suffer from catastrophic forgetting and are often not suitable for long task sequences (Kirkpatrick et al., 2017; Razdaibiedina et al., 2022). In contrast to regularization-based and replay-based approaches, architectural CL approaches are more efficient in resolving catastrophic forgetting and, hence, are suitable for long sequences of tasks.

**Parameter-efficient Learning** Recent works on parameter-efficient learning have shown that by training a subset of parameters, we can achieve a full model performance (Houlsby et al., 2019; Li & Liang, 2021; Karimi Mahabadi et al., 2021; Lester et al., 2021). While this line of work has mostly focused on learning a single task, there have been some attempts on using parameter-efficient tuning for CL (Madotto et al., 2020; Qin & Joty, 2021; Wang et al., 2022b;a). For instance, Madotto et al. (2020) proposes AdapterCL which learns a separate adapter block for each task, and Qin & Joty (2021) proposes LFPT5 that learns a large (length 300) soft prompt that is continuously trained on all tasks. While AdapterCL resolves catastrophic forgetting problem, it does not allow forward transfer. In contrast, LFPT5 allows forward transfer but suffers from forgetting.

## 7 CONCLUSION

This paper presents Progressive Prompts – a novel approach for CL that addresses catastrophic forgetting in pre-trained language models, while allowing knowledge reuse from previous tasks. In contrast to many existing CL methods for NLP, the proposed approach does not require saving examples from the previous tasks for data replay. Moreover, our method does not require storing a large number of task-specific parameters. Progressive Prompts is a model-agnostic approach and our experiments with two commonly used language models demonstrate that Progressive Prompts outperforms baseline methods on a standard CL benchmark for text classification and our custom benchmark of longer CL sequences that spans 15 tasks.

ACKNOWLEDGMENTS

We thank Jimmy Ba for many fruitful discussions and brainstorming of the ideas, which helped to improve this project. We also thank Victoria Lin and the FAIR team for their helpful comments. Finally, we thank the anonymous reviewers whose feedback helped to improve this paper.

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

# Appendix

## A   FURTHER IMPLEMENTATION DETAILS

### A.1   DATASETS

Table 5 shows details of the 15 datasets we used for our CL experiments, along with their evaluation metrics. Overall, we used datasets from CL benchmark (Zhang et al., 2015), GLUE (Wang et al., 2018) and SuperGLUE (Wang et al., 2019) benchmarks, and added IMDB movie reviews dataset. Following common practive, for tasks that have two evaluation metrics we use the average of the two as the final performance metric.

| Dataset name | Category | Task | Domain | Metric |
|---|---|---|---|---|
| 1. Yelp | CL benchmark | sentiment analysis | Yelp reviews | accuracy |
| 2. Amazon | CL benchmark | sentiment analysis | Amazon reviews | accuracy |
| 3. DBpedia | CL benchmark | topic classification | Wikipedia | accuracy |
| 4. Yahoo | CL benchmark | QA | Yahoo Q&A | accuracy |
| 5. AG News | CL benchmark | topic classification | news | accuracy |
| 6. MNLI | GLUE | NLI | various | accuracy |
| 7. QQP | GLUE | paraphrase detection | Quora | accuracy & F1 |
| 8. RTE | GLUE | NLI | news, Wikipedia | accuracy |
| 9. SST2 | GLUE | sentiment analysis | movie reviews | accuracy |
| 10. WiC | SuperGLUE | word sense disambiguation | lexical databases | accuracy |
| 11. CB | SuperGLUE | NLI | various | accuracy |
| 12. COPA | SuperGLUE | QA | blogs, encyclopedia | accuracy |
| 13. BoolQ | SuperGLUE | boolean QA | Wikipedia | accuracy |
| 14. MultiRC | SuperGLUE | QA | various | F1 & EM |
| 15. IMDB | Other | sentiment analysis | movie reviews | accuracy |

Table 5: The details of 15 datasets used in our CL experiments. NLI denotes natural language inference, QA denotes questions and answers task, EM denotes exact match scoring. First five tasks correspond to the standard CL benchmark, all other tasks are used in our long-sequence experiments.

### A.2   TASK SEQUENCE ORDERS

We report task orders used for our CL experiments across BERT and T5 models in Table 6 below:

| Order | Model | Task Sequence |
|---|---|---|
| 1 | T5 | db → amazon → yahoo → ag |
| 2 | T5 | db → amazon → ag → yahoo |
| 3 | T5 | yahoo → amazon → ag → db |
| 4 | BERT | ag → yelp → amazon → yahoo → db |
| 5 | BERT | yelp → yahoo → amazon → db → ag |
| 6 | BERT | db → yahoo → ag → amazon → yelp |
| 7 | BERT | yelp → ag → db → amazon → yahoo |
| 8 | T5, BERT | mnli → cb → wic → copa → qqp → boolq → rte → imdb → yelp → amazon → sst2 → dbpedia → ag → multirc → yahoo |
| 9 | T5, BERT | multirc → boolq → wic → mnli → cb → copa → qqp → rte → imdb → sst2 → dbpedia → ag → yelp → amazon → yahoo |
| 10 | T5, BERT | yelp → amazon → mnli → cb → copa → qqp → rte → imdb → sst2 → dbpedia → ag → yahoo → multirc → boolq → wic |

Table 6: Ten different orders of task sequences used for continual learning experiments. Orders 1-7 correspond to the standard CL becnhmark adopted by prior works. Orders 8-10 are our custom long-sequence orders spanning 15 tasks.

### A.3 IMPLEMENTATION

We use PyTorch (Paszke et al., 2019) and HuggingFace Transformers library (Wolf et al., 2019) for our implementation. For the standard CL benchmark, we use official datasets provided by Zhang et al. (2015) available at `http://goo.gl/JyCnZq`, following de Masson D'Autume et al. (2019); Zhang et al. (2015). We use HuggingFace datasets (`https://github.com/huggingface/datasets`) to download data for GLUE tasks (Wang et al., 2018), SuperGLUE tasks (Wang et al., 2019) tasks, and IMDB movie reviews dataset (Maas et al., 2011), which we use for long-sequence CL experiments and/or ablation studies. Following previous studies (Rao et al., 2019; de Masson D'Autume et al., 2019), for CL experiments, for each dataset we use the available validation set as a test set (since test data is not available), and hold out 500 samples from the train set to construct the validation set. For our ablation studies, since we compare Progressive Prompts with the original prompt tuning (Lester et al., 2021), we follow their set up and report maximal validation set performance.

### A.4 EXPERIMENT DETAILS

We use Adam optimizer (Kingma & Ba, 2014) and set batch size to 8 for all the experiments, except for MTL runs with a batch size of 2 (due to memory limitations). We train each prompt between 10 and 300 epochs, depending on the number of data points. We use the prompt checkpoints with the best validation set score as our final prompts. Prompts are initialized from randomly sampled tokens as in Lester et al. (2021), hyperparametes are shown in the Table 7 below:

| Hyperparameter ↓ | CL benchmark | Long-sequence benchmark | | |
|---|---|---|---|---|
| Num. samples → | - | 1000 | 200 | 20 |
| **BERT** | | | | |
| Epochs | 40 | 40 | 150 | 300 |
| Learning rate | $1e-4$ | $1e-4$ | $1e-4$ | $1e-4$ |
| Prompt length | 20 | 20 | 20 | 20 |
| **T5** | | | | |
| Epochs | 10 | 10 | 150 | 300 |
| Learning rate | 0.3 | 0.3 | 0.3 | 0.3 |
| Prompt length | 50 | 10 | 10 | 10 |

Table 7: Hyperparameters used for Progressive Prompts across different CL experiments.

For all CL experiments we use early stopping as in Huang et al. (2021), to save model checkpoint based on the best validation performance on the current task. We report test set performance after training on all tasks as our final metric. For SuperGLUE experiments, we report maximal validation set performance over the course of training as in Lester et al. (2021). We measure the validation performance after every epoch and use metrics described in Appendix A.1. We use 1% of samples per class for the replay approach (but no less than 1 sample per class), following Huang et al. (2021). We use the same hyperparameter setting for all prompt-based approaches (Progressive Prompts, prompt tuning, per-task prompts), except for prompt tuning we use a longer shared prompt of 200 tokens. For all other approaches, we use hyperparameters provided in their corresponding papers.

## B FURTHER EXPERIMENTAL RESULTS

### B.1 LONG SEQUENCE EXPERIMENTS

Here we report results for orders 8, 9 and 10 of continual learning experiments with long task sequences of 15 tasks. We show test set performance averaged across all tasks for each method. Test scores are calculated after training has been completed. Results are shown in Table 9. We also investigate improvement of Progressive Prompts compared to per-task prompt on the corresponding task in Figure 12. Clearly, some tasks benefit from knowledge sharing from the progressively added prompts. Additionally, we assessed if initializing

| Method | Few-shot | Full-shot |
|---|---|---|
| PT + Prev. Init. | 48.2 | 50.0 |
| ProgPrompt | **53.5** | **69.3** |

Table 8: Comparison of Prompt Tuning and Progressive Prompts, when shared promot in Prompt Tuning is initialized from the previous task. Average test accuracy after observing all tasks is shown (averaged across three orders).

new prompt from the previous task prompt would result in better performance of continual prompt tuning. We observe that Progressive Prompts outperform this setup in both few-shot (20/class) and full-shot settings, see Table 8.

In addition to average performance, we compute more fine-grained performance metrics defined by Lopez-Paz & Ranzato (2017) for different approaches under long-sequence experiments. Specifically, we compute *backward transfer* and *forward transfer* metrics, and *evolution of average accuracy* over learning new tasks. Our results on FWT for task orders 8, 9 and 10 are shown in Figure 5, Figure 6 and Figure 7 respectively. Our results on BWT for task orders 8, 9 and 10 are shown in Figure 8, Figure 9 and Figure 10 respectively. Evolution of accuracies is shown in Figure 11.

| Method ↓ | Order 8 | | | Order 9 | | | Order 10 | | | avg | | |
| Num. samples → | 20 | 200 | 1000 | 20 | 200 | 1000 | 20 | 200 | 1000 | 20 | 200 | 1000 |
|---|---|---|---|---|---|---|---|---|---|---|---|---|
| **T5-Large results** | | | | | | | | | | | | |
| Finetune | 9.3 | 8.9 | 7.4 | 9.5 | 8.1 | 7.4 | 10.4 | 7.9 | 7.5 | 9.7 | 8.3 | 7.4 |
| Replay | 46.0 | 45.0 | 55.2 | 50.3 | 43.7 | 54.8 | 34.6 | 43.8 | 53.3 | 43.6 | 44.2 | 54.4 |
| PromptTuning | 9.7 | 8.4 | 8.2 | 24.4 | 16.8 | 8.7 | 12.2 | 8.0 | 7.9 | 17.4 | 13.9 | 10.9 |
| Per-task Prompts | 69.9 | 75.2 | 77.0 | 69.9 | 75.2 | 77.0 | 69.9 | 75.2 | 77.0 | 69.8 | 75.2 | 77.0 |
| LFPT5 | 54.7 | 61.6 | 70.4 | 54.1 | 54.3 | 68.2 | 54.2 | 58.8 | 69.1 | 54.3 | 58.2 | 69.2 |
| ProgPrompt | **75.4** | **79.1** | **79.5** | **76.6** | **78.2** | **79.1** | **76.7** | **78.9** | **79.8** | **76.2** | **78.7** | **79.5** |
| MTL | 70.7 | 72.5 | 76.3 | 70.7 | 72.5 | 76.3 | 70.7 | 72.5 | 76.3 | 70.7 | 72.5 | 76.3 |
| **BERT-base results** | | | | | | | | | | | | |
| Finetune | 29.9 | 43.4 | 40.9 | 30.5 | 42.0 | 42.5 | 33.6 | 41.9 | 41.8 | 31.3 | 42.4 | 41.7 |
| Replay | 34.9 | 46.3 | 51.0 | 39.3 | 48.1 | 51.5 | 34.9 | 46.5 | 46.3 | 36.4 | 47.0 | 49.6 |
| Per-task Prompts | 50.6 | 62.4 | 67.2 | 50.6 | 62.4 | 67.2 | 50.6 | 62.4 | 67.2 | 50.6 | 62.4 | 67.2 |
| IDBR | 39.7 | 48.4 | 52.3 | 37.9 | 46.6 | 54.1 | 32.9 | 48.8 | 50.1 | 36.8 | 47.9 | 52.2 |
| ProgPrompt | **55.3** | **67.9** | **68.9** | **53.3** | **65.8** | **70.0** | **51.9** | **66.9** | **69.0** | **53.5** | **66.9** | **69.3** |
| MTL | 56.9 | 67.7 | 69.9 | 56.9 | 67.7 | 69.9 | 56.9 | 67.7 | 69.9 | 56.9 | 67.7 | 69.9 |

Table 9: Average test set performance of Progressive Prompts and common CL approaches on long-sequence experiments with 15 text classicication tasks (orders 8, 9 and 10). We report results for BERT and T5 models across different limits of data – 20, 200 and 1000 samples per class. MTL denotes multi-task learning. All results are averaged over 3 runs.

## B.2 SUPERGLUE EXPERIMENTS SETUP

Comparison of the original prompt tuning on SuperGLUE (Lester et al., 2021) and our Progressive Prompt setup is shown in Figure 13.

## B.3 EFFECT OF PROMPT REPARAMETERIZATION

We find that residual reparameterization allows to reach performance close to finetuning, and is especially helpful for BERT model. Table 10 shows the results of regular prompt tuning, prompt tuning with MLP reparameterization and prompt tuning with residual MLP reparameterization on BERT-base model. As in all our experiments, we use a 2-layer MLP with the hidden layer dimension of 800. We show the best performance on four different tasks with prompts of length 5 and 30. Following Lester et al. (2021), we report the maximal validation set performance for each dataset. Our results show that residual MLP reparameterization results in performance improvement over standard prompt tuning, reaching close to finetuning performance (Table 10). Notably, with length-5 prompt, residual MLP improves accuracy by approximately 6% and 4% for IMDB and QQP datasets, matching full model tuning. Regular MLP reparameterization generally leads to either smaller improvement than residual MLP or even worse performance than prompt tuning.

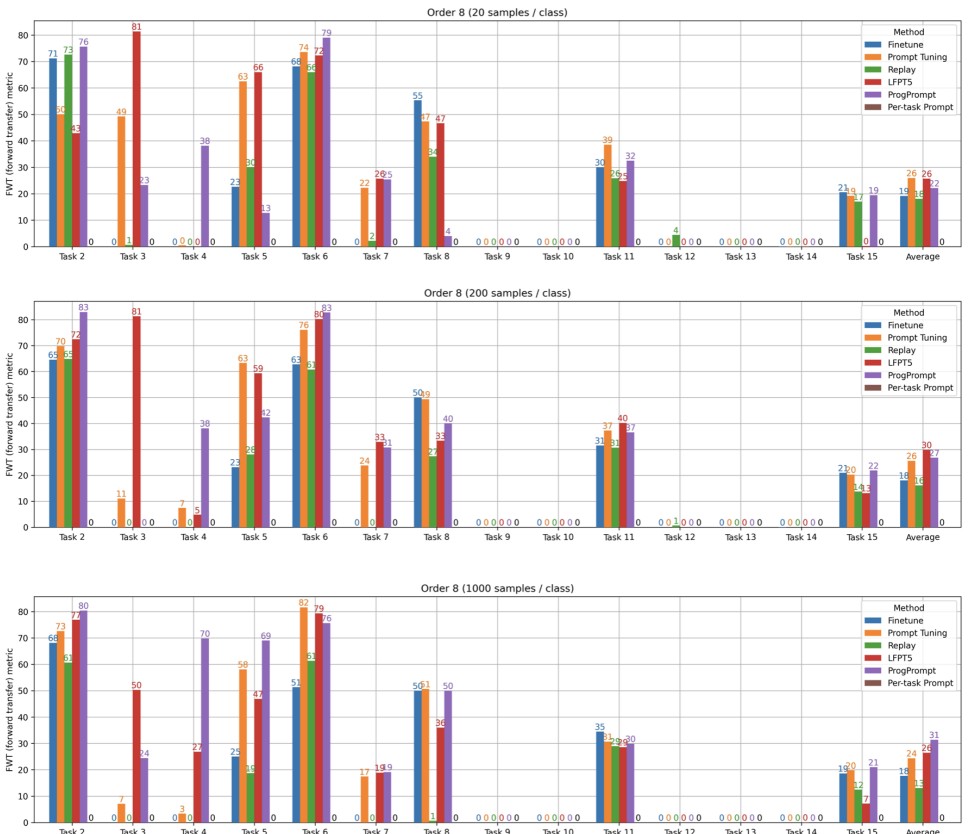

Figure 5: Forward transfer score of different approaches on order 8. Different data limits are shown (20, 200 and 1000 samples per class).

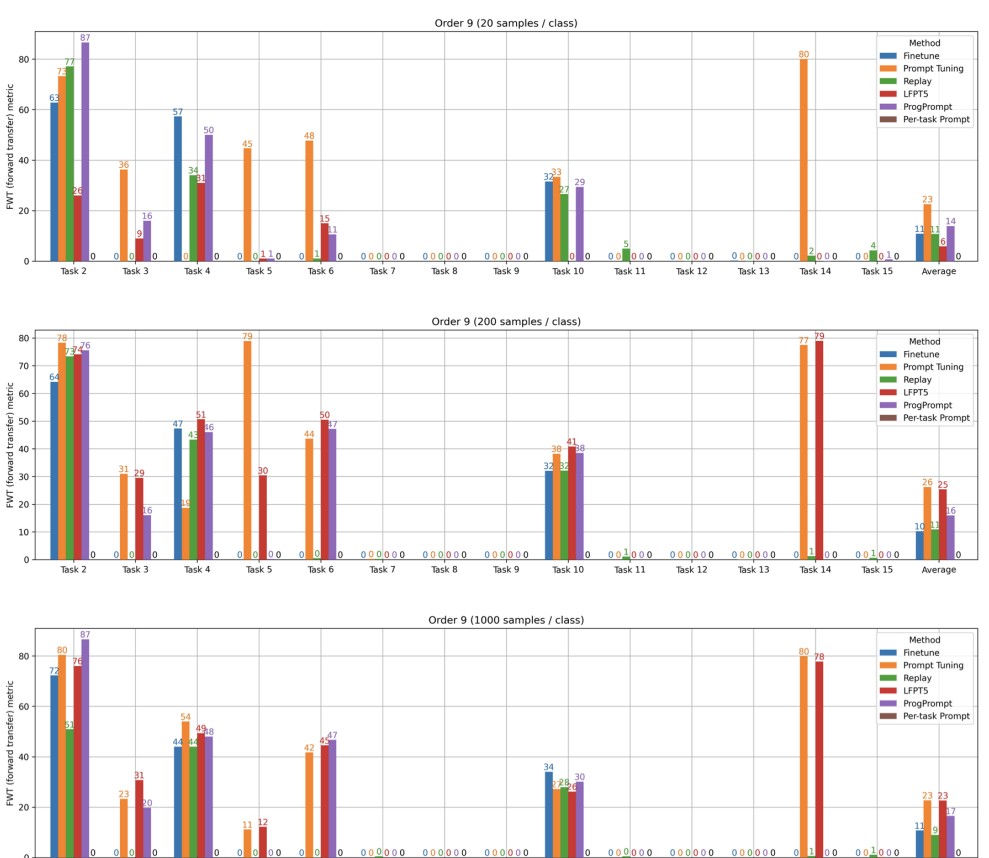

Figure 6: Forward transfer score of different approaches on order 9. Different data limits are shown (20, 200 and 1000 samples per class).

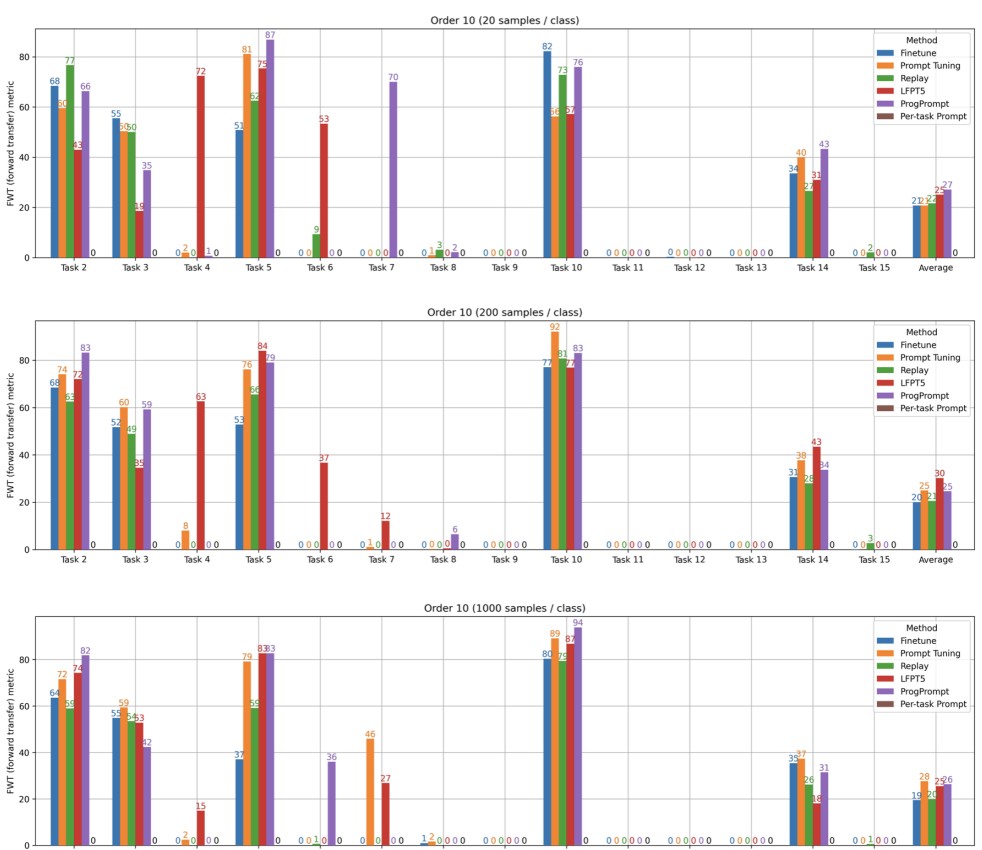

Figure 7: Forward transfer score of different approaches on order 10. Different data limits are shown (20, 200 and 1000 samples per class).

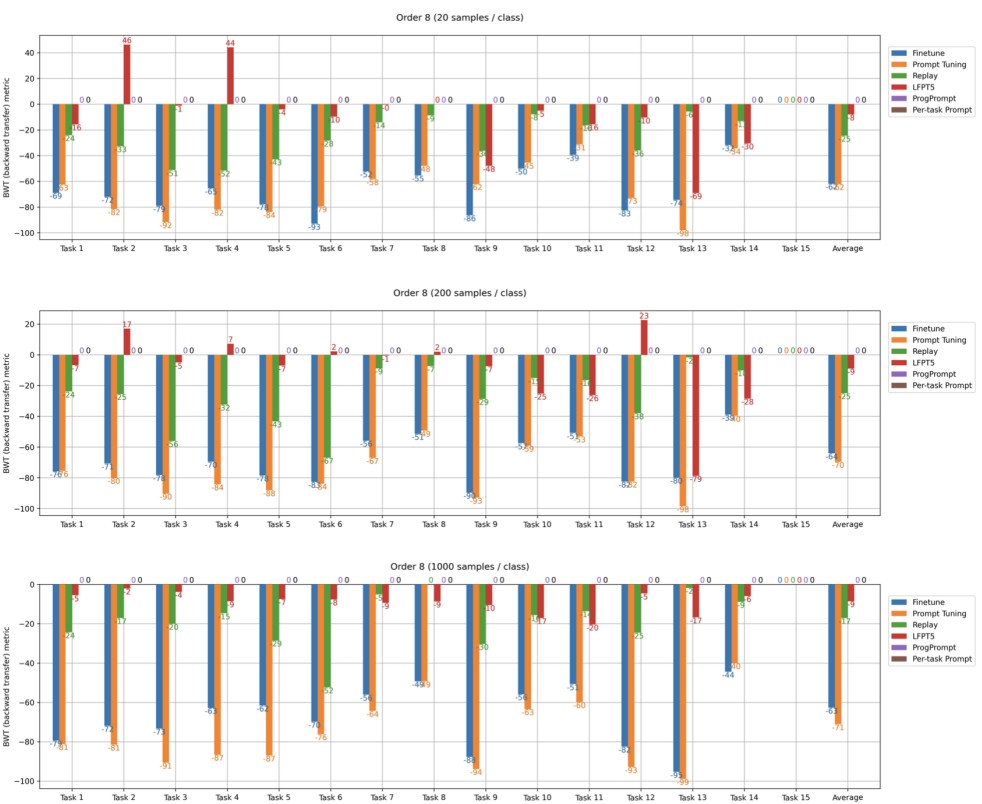

Figure 8: Backward transfer score of different approaches on order 8. Different data limits are shown (20, 200 and 1000 samples per class).

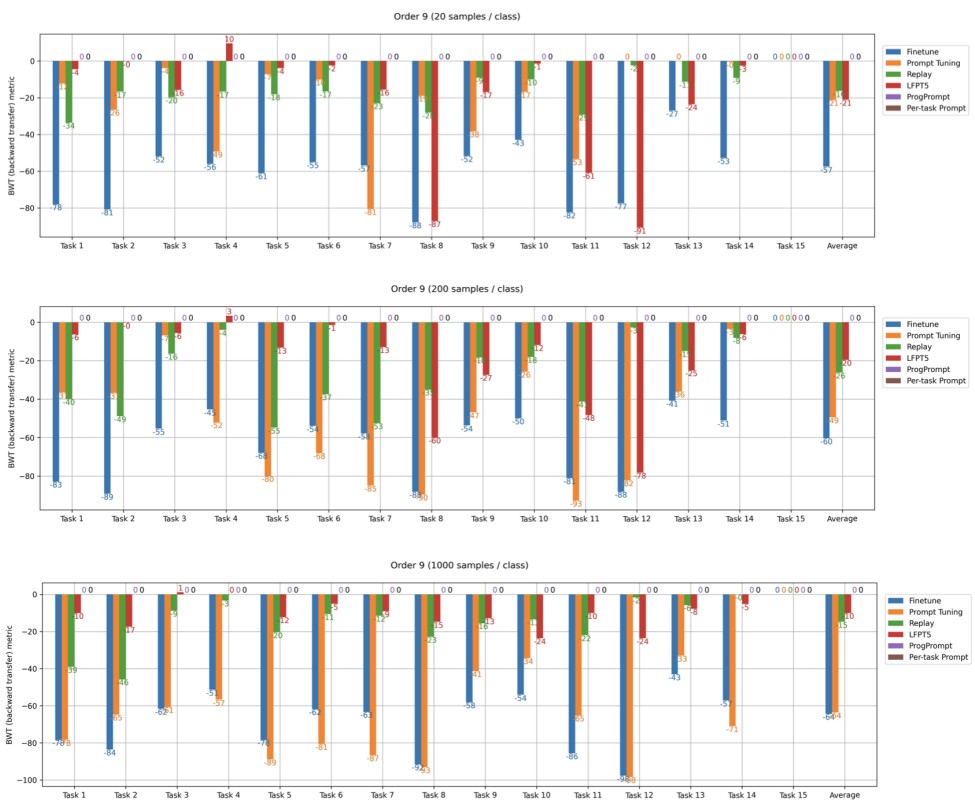

Figure 9: Backward transfer score of different approaches on order 9. Different data limits are shown (20, 200 and 1000 samples per class).

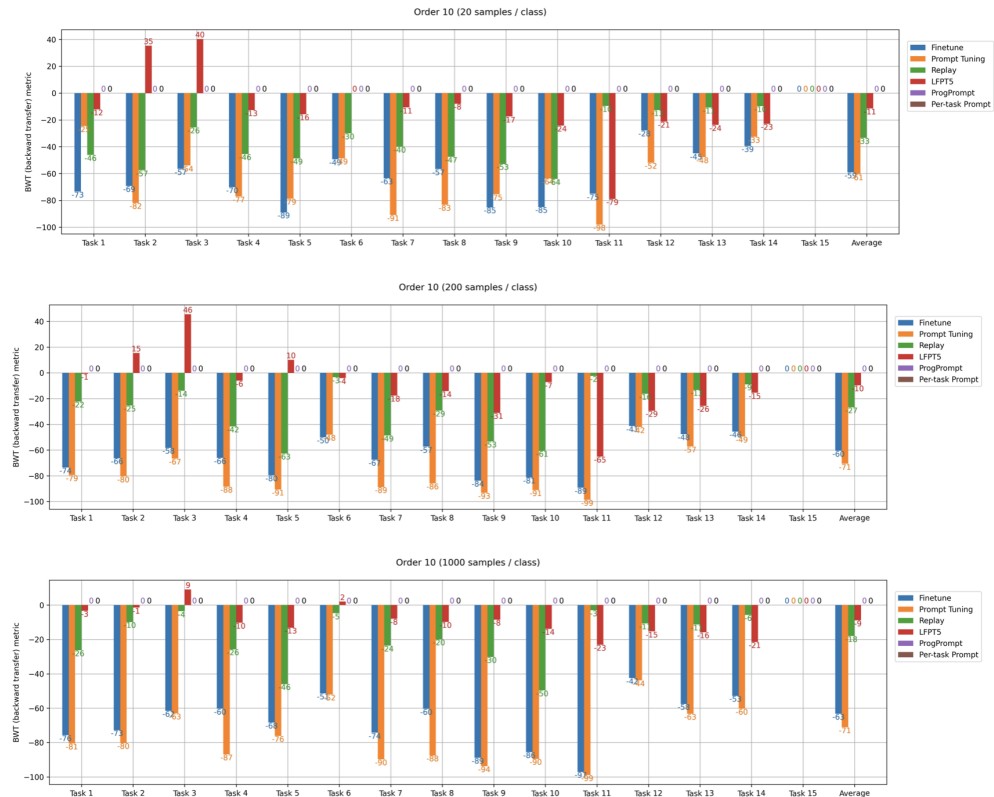

Figure 10: Backward transfer score of different approaches on order 10. Different data limits are shown (20, 200 and 1000 samples per class).

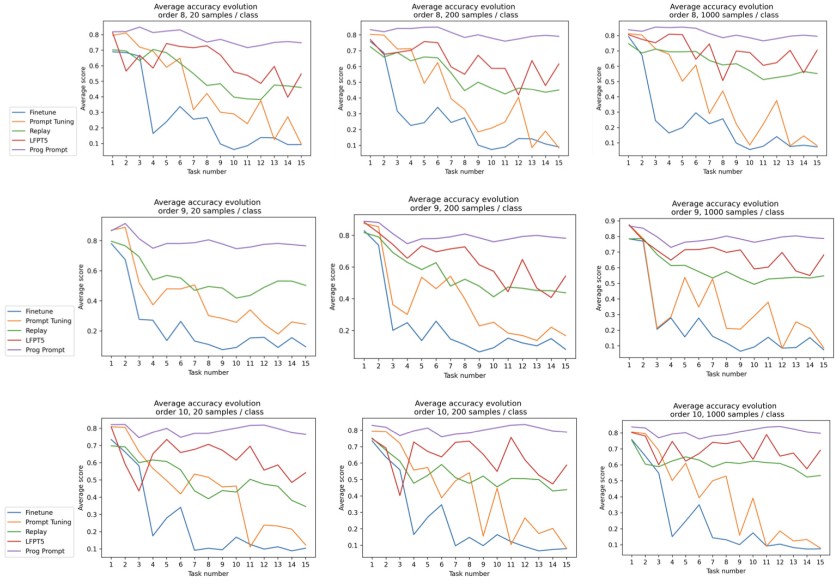

Figure 11: Evolution of average accuracy after learning new tasks.

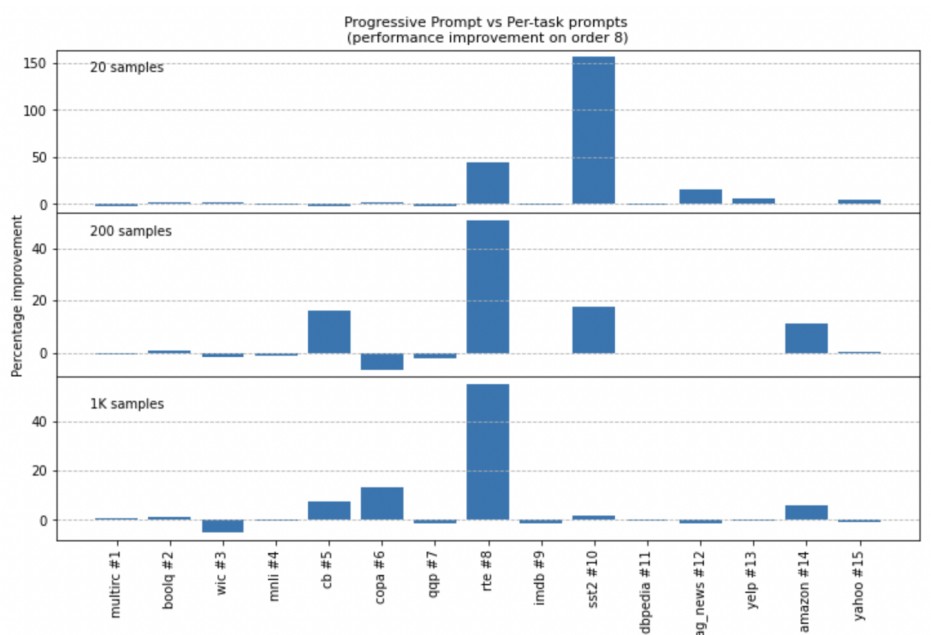

Figure 12: Per-task improvement of Progressive Prompts verus per-task prompts in CL experiment with order 8 across different data limits (20, 200 and 1000 samples per class). X-axis shows the sequence of tasks, Y-axis shows percentage improvement of Progressive Prompts test score compared to per-task prompt on the corresponding task.

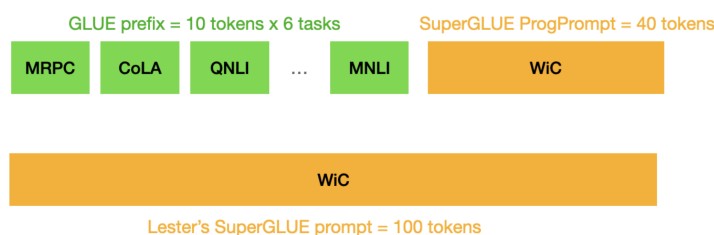

Figure 13: Original prompt tuning versus Progressive Prompts on SuperGLUE datasets. For illustration, we show how SuperGLUE task WiC is leaned (we have similar scheme for other tasks). Prompt tuning trains a single prompt of 100 tokens for WiC task. Progressive Prompts method learns a prompt of 40 tokens, which is progressively appended to the six frozen prompts of 10 tokens learned on GLUE benchmark (with random task order). Total prompt length is equal in both approaches.

| Prompt len. $\rightarrow$ | | length 5 | | | length 30 | | |
| Task $\downarrow$ | PT | PT+MLP | PT+resMLP | PT | PT+MLP | PT+resMLP | FT |
|---|---|---|---|---|---|---|---|
| IMDB | $85.8_{0.7}$ | $83.4_{0.9}$ | $\mathbf{91.4_{0.8}}$ | $89.1_{1.0}$ | $90.2_{1.5}$ | $\mathbf{91.2_{1.4}}$ | $92.9_{0.4}$ |
| QQP | $72.9_{0.8}$ | $73.1_{1.1}$ | $\mathbf{76.6_{1.2}}$ | $72.2_{1.5}$ | $73.9_{4.1}$ | $\mathbf{77.3_{0.9}}$ | $78.6_{0.9}$ |
| RTE | $\mathbf{65.0_{2.3}}$ | $64.8_{3.2}$ | $\mathbf{65.0_{3.5}}$ | $\mathbf{67.8_{3.2}}$ | $67.6_{3.6}$ | $66.2_{3.1}$ | $66.2_{2.0}$ |
| MRPC | $72.1_{0.8}$ | $71.5_{1.4}$ | $\mathbf{75.8_{0.9}}$ | $73.4_{0.8}$ | $76.9_{3.4}$ | $\mathbf{79.0_{1.3}}$ | $86.3_{1.6}$ |

Table 10: Effect of prompt embeddings reparametrization on prompt tuning performance with BERT model. Average performance across 3 runs is shown. **PT**: regular prompt tuning, **PT+MLP**: prompt tuning with prompt passed through 2-layer MLP, **PT+resMLP** (our approach): prompt tuning with prompt passed through 2-layer MLP with a skip connection (*residual MLP*). **FT**: full model finetuning. Best results for each prompt length (5 and 30) are highlighted in bold.

