# OpenReview forum: "Progressive Prompts: Continual Learning for Language Models"
_ICLR.cc/2023/Conference — ICLR 2023 poster_

### Official Review · Reviewer_KFhT · 2022-10-23

**Confidence:** 4
**Correctness:** 4
**Technical Novelty And Significance:** 2
**Empirical Novelty And Significance:** 3
**Recommendation:** 6

**Clarity, Quality, Novelty And Reproducibility:**


**Clarity & Quality:** Overall, the paper is well-written. However, I find the methodology section very brief and a few important details missing.


**Novelty**: The majority of the fundamental ideas, such as prompt tuning, progressive training, and embedding reparametrization, are borrowed from previous works. However, I do not find this a major concern.


**Reproducibility**: The authors explained most of their experimental setup.




**Strength And Weaknesses:**

**Strengths**


- The paper is well-written and easy to follow. The idea of prompt tuning is also intuitive.


- The proposed method shows strong performance.




**Weaknesses**


- One major of the work is the assumption of accessing task identifiers (for separating prompts and task parameters). I believe this assumption (similar to the majority of task-incremental works), is very relaxed for practical scenarios. For example, task boundaries and identifiers do not always exist in real-world streaming setups.



- In addition, although very small, the computational costs grow as more and more tasks arrive. This is also the case for classical expansion-based CL algorithms and, at some point (e.g., hundreds of tasks), not negligible.



- The main focus of the paper is on the final average performance (e.g., accuracy, F1) of the models/algorithms. However, it is important to compare continual learning methods from different perspectives that measure the evolution of the accuracies and forward and backward transfers.




**Questions**


- Could the authors provide a more fine-grained analysis of the methods on at least one benchmark (e.g., with many tasks)? For instance, how the average accuracy evolves as each task is learned, how much forgetting (if any) we have, etc. ?


- Does ProgPrompt require task identifiers at the test time?


- Could the author provide more fine-grained performance measures


- As explained in Section 3, ProgPrompt allocates different prompts and parameters for each task. Could the authors provide the total parameters/time/computation comparison of ProgPrompt vs. other methods for the experiments with a large number of tasks?





**Summary Of The Paper:**

The paper studies continual learning in the NLP domain, focusing on pre-trained language models with transformers. The authors use various techniques to improve the continual learning performance, such as prompt tuning and embedding regularization. The proposed method, Progressive Prompt (ProgPrompt), learns a prompt for each new task. In addition, the base language model is kept frozen while a small number of parameters is added for each task. Since prompts and per-task expanded parameters are separate, this allows for reducing catastrophic forgetting. The paper then empirically shows the effectiveness of ProgPrompt for BERT and T5 models on various NLP benchmarks.

**Summary Of The Review:**

**Initial**
Overall, while I find the proposed method intuitive with strong results, I believe many details are missing from the paper. In addition, I find the assumption of accessing task-identifiers very relaxed. While I mentioned from the novelty perspective, the work heavily relies on previous ideas, I do not find this a major limitation.

**Update**
I would like to thank the authors for their response. While the proposed method still has a limitation regarding the task-boundary assumption, given the clarifications and new results, I increased the score, and I encourage the authors to add the full result to the paper.

---

> ### Author Response · Authors · 2022-11-19
> **Response to Reviewer KFhT**
>
> Thank you for your detailed review of our paper and constructive feedback! We are encouraged that you found our idea intuitive, our method effective, and our writing clear. We updated our paper with the suggested additional metrics, and we address your comments below:
>
> **Q1: “Could the authors provide a more fine-grained analysis of the methods on at least one benchmark (e.g., with many tasks)?”, “... measure the evolution of the accuracies and forward and backward transfers.”**
>
> **A1**: Thank you for the suggestion! Here we *provide 3 additional metrics* on our long-sequence benchmark: *average accuracy evolution*, *backward transfer (BWT)* and *forward transfer (FWT)* as defined by Lopez-Paz et al. (NeurIPS, 2017).
>
> Here is a short summary of our results on 15-task experiments with T5 model:
>
> 1. ***Backward transfer*** (higher is better, 0 = no forgetting)
>
> The table below shows average BWT across 3 task orders of long-sequence experiments:
>
> | Method           | Few-shot (20/class) | Full-shot |
> |------------------|:-------------------:|:---------:|
> | Finetune         | -59.5               | -63.5     |
> | Replay           | -24.7               | -18.0     |
> | Prompt Tuning    | -47.9               | -71.0     |
> | Per-task Prompts | **0**               | **0**     |
> | LFPT5            | -13.5               | -8.8      |
> | ProgPrompts      | **0**               | **0**     |
>
> Full BWT results on all tasks across 3 orders in N-shot settings (N={20, 200, 1000}) are available in Appendix A.5 and also here:
> https://drive.google.com/drive/folders/1kk759Lbi6a847k-lgeO_L_Qrohovwi8r?usp=sharing
>
> 2. ***Forward transfer*** (higher is better)
>
> The table below shows average FWT across 3 task orders of long-sequence experiments:
>
> |   Method         | Few-shot (20/class) | Full-shot |
> |------------------|:-------------------:|:---------:|
> | Finetune         | 16.9                | 16.0      |
> | Replay           | 16.8                | 14.0      |
> | Prompt Tuning    | **23.1**            | **24.9**  |
> | Per-task Prompts | 0                   | 0         |
> | LFPT5            | 18.9                | **24.9**  |
> | ProgPrompts      | _21.1_              | _24.7_    |
>
> Full FWT results on all tasks across 3 orders in N-shot settings (N={20, 200, 1000}) are available in Appendix A.5 and also here:
> https://drive.google.com/drive/folders/1OFOIzrnoC48kTC6clKcYW9CiMt5A0uuH?usp=sharing
>
> 3. ***Evolution of accuracies*** are shown on all tasks across 3 orders in Appendix A.5 and also here:
> https://drive.google.com/file/d/1QodB7nt3-fp-pqKqPcIEhqtWK2DUC3Me/view?usp=share_link
>
> We added all these results in Appendix A.5 in our updated version of the paper.
>
> *Result summary*: We observe that our method, **Progressive Prompts, achieves the best BWT score and second-best FWT score** in both few-shot and full-shot continual learning setup. Notably, even though Progressive Prompts do not require any data replay - they score on par in FWT with replay-based methods (0.2% difference). Also, Progressive Prompts **consistently achieve the highest average accuracy** as it evolves during learning new tasks.
>
>
> **Q2: “Does ProgPrompt require task identifiers at the test time?”**
>
> **A2**: Yes - we followed the continual learning protocol by previous state-of-the art method LFPT5 (Qin et al., ICLR 2022), which allows use of task identifiers, and improved the performance over existing approaches. However, we agree that in our future work it would be interesting to expand Progressive Prompts for usage without task identifiers. E.g. we could train another network to infer the task identity at the test time.
>
>
> **Q3: “Could the authors provide the total parameters/time/computation comparison of ProgPrompt vs. other methods for the experiments with a large number of tasks?”**
>
> **A3**:  Again thanks for your suggestion! We provide analysis on parameter-efficiency (estimations for long-sequence experiments on T5, num. trainable parameters):
>
> |                     | Params / task | Total params  |
> |---------------------|---------------|---------------|
> | Finetune            | 770M          | 770M          |
> | Replay              | 770M          | 770M          |
> | Prompt Tuning       | 205K          | 205K          |
> | Per-task Prompts    | **10K**       | **154K**      |
> | LFPT5               | 307K          | 307K          |
> | ProgPrompts w/ MLP  | 720K          | 864K          |
> | ProgPrompts w/o MLP | **10K**       | **154K**      |
>
> Overall, even though Progressive Prompts grow in size as more tasks arrive, the expenses are very small. **Progressive Prompts** (without MLP reparameterizion) and per-task prompts are **the most parameter-efficient** approaches among explored methods.
> In our T5 model experiments, we add a prompt of only 10 tokens per task, which in the longest task sequence would add 150 tokens. Compared to LFPT5, which uses a single prompt but with 300 tokens, our total Progressive Prompt sequence is more efficient.

---

> ### Author Response · Authors · 2022-12-03
> **Thank you, Reviewer KFhT!**
>
> Thank you very much for the update! We are happy that you found our new results / clarifications useful! Your suggestions helped to improve our paper a lot. We already included per-task results to the Appendix, and will also include tables with average metrics to the final version (as you suggested).

---

### Official Review · Reviewer_Dpx6 · 2022-10-26

**Confidence:** 4
**Correctness:** 4
**Technical Novelty And Significance:** 3
**Empirical Novelty And Significance:** 3
**Recommendation:** 8

**Clarity, Quality, Novelty And Reproducibility:**

Clarity
The paper is overall well written

Novelty
The method proposed in the paper is novel, it is a combination of exisiting ideas but the current formulation reduces the memory footprint by a big margin.

Reproducibility
There is no mention of code release by the authors.

**Strength And Weaknesses:**

Strength
The method proposed in the paper is an efficient way to continually learn language tasks, requiring to train a very small number of parameters as compared to the overall model sizes (< 0.1%). The method is quite simple and intuitive, and is explained quite well in the paper.

Weakness
There are a couple of questions that the paper raises:
- Section 2.3 mentions the framework that the identity of each task is known a priori. Instead of concatenating the prompt vectors over time, how would the following baseline perform: the prompt has only one theta_p. update \theta_p for each task starting from the previous one as initialization? If this alrternative is nearly as good, this might save a lot fof time at inference.
- Another alternative is to directly train a prompt of size m \times |\theta_p| instead of one sequential vector per task. How would this perform as a baseline?
- For the results in Section 5, can you also provide numbers for per task fine tuning as a baseline? This would help calibrate how much transfer can you actually get from other tasks.
- In section 5.4, the training procedure to obtain the numbers in Table 4 is modified from the one described before. Why is this the case?

**Summary Of The Paper:**

This paper proposes a new prompting based method for continual learning for language models. The method learns a sequnce of task specific prompts which allows it to overcome the forgetting problem for long sequences. The methods achieve state of the art performance on long range continual learning tasks.

**Summary Of The Review:**

The paper proposes a new method for continual learning which is intereting. I believe some baseline comparisons are actually missing from their experiment setup which have been pointed out in the weakness section.

---

> ### Author Response · Authors · 2022-11-19
> **Response to Reviewer Dpx6**
>
> Thank you for your detailed review of our paper and constructive feedback! We are encouraged that you found our idea promising, our method effective and novel, and our writing clear. We are addressing each of your comments / questions below:
>
> **Q1: “Instead of concatenating the prompt vectors over time, how would the following baseline perform: the prompt has only one $\theta_p$. update $\theta_p$ for each task starting from the previous one as initialization?”**
>
> **A1**: Thank you for the suggestion! We conducted new experiments with the suggested setup. Here are the results for a long sequence of tasks for T5 model (average test accuracy after training on all tasks is shown):
> |                      | Few-shot (20/class) | Full-shot |
> |----------------------|---------------------|-----------|
> | Prompt Tuning + Init | 48.2                | 60.0      |
> | Progressive Prompt   | **53.5**            | **69.3**  |
>
> Overall, updating $\theta_p$ from the previous task initialization results in lower performance than Progressive Prompts, which could be explained by forgetting.
>
> We have included these experiments and baselines in our latest revision (Appendix A.5).
>
>
> **Q2: “Another alternative is to directly train a prompt of size $m \times |\theta_p|$ instead of one sequential vector per task. How would this perform as a baseline?”**
>
> **A2**: Thank you for the suggestion! We already include this baseline as “Prompt Tuning” in Table 2. For Prompt Tuning, we continuously train a large prompt of 200 tokens (instead of 15 separate prompts of 10 tokens each as in Progressive Prompts). Here is a short version of the results for T5 model (average test accuracy across all tasks in three orders is shown):
>
> |                    | Few-shot (20/class) | Full-shot |
> |--------------------|---------------------|-----------|
> | Prompt Tuning      | 47.6                | 59.5      |
> | Progressive Prompt | **53.5**            | **69.3**  |
>
> We observe that a large shared prompt performs worse than a collection of smaller-sized prompts (due to forgetting that happens on long sequences of tasks). Therefore, a prompt of size $m \times |\theta_p|$ is less suitable for long-sequence continual learning compared to our approach.
>
>
> **Q3: “For the results in Section 5, can you also provide numbers for per task fine tuning as a baseline? This would help calibrate how much transfer can you actually get from other tasks.”**
>
> **A3**: We updated our latest revision  with per-task fine-tuning results in Sections 5.1 and 5.2. Here is a summary of the results with *long-sequence experiments* (averaged test accuracy across all tasks and three orders, with T5 model):
>
> |                      | Few-shot (20/class) | Full-shot |
> |----------------------|---------------------|-----------|
> | Per-task fine-tuning | 68.2                | 78.1      |
> | Progressive Prompt   | **76.2**            | **79.5**  |
>
> And a summary for the *standard continual learning benchmarks* (here we used per-task fine-tuning accuracies reported by Qin et al., ICLR 2022 and Huang et al., NAACL 2021):
> |                      | BERT benchmark | T5 benchmark |
> |----------------------|----------------|--------------|
> | Per-task fine-tuning | 73.9           | 70.0         |
> | Progressive Prompt   | **77.9**       | **75.1**     |
>
> Full tables are available in the revised paper, Sections 5.1 and 5.2, we also include the screenshots here:
> * Standard benchmarks on BERT and T5 (4 and 5 tasks): https://drive.google.com/file/d/1xuJYv1oJnou77KH1_c4BDtq7evlLN6rj/view?usp=sharing
> * Long-sequence continual learning (15 tasks): https://drive.google.com/file/d/1x1I4WRqs7yVK3fGHZgk8K-O-BAoMZ98f/view?usp=share_link
>
> Overall, we observe most transfer with Progressive Prompts in a few-shot setup. Specifically, in  long-sequence continual learning experiments Progressive Prompts achieve +8% over per-task fine-tuning in few-shot setup (20 samples/class), and +1.4% improvement in full-shot setup.
>
>
> **Q4: “In section 5.4, the training procedure to obtain the numbers in Table 4 is modified from the one described before.”**
>
> **A4**: Since experiments in Table 4 are comparing our method to prompt tuning, we followed the original Prompt Tuning paper setup by Lester et al. to assess performances of our approaches under the same conditions. The rest of our experiments were performed in a continual learning setting and, hence, we followed the standard continual learning protocol used in previous studies (Qin et al., ICLR 2022; Huang et al., NAACL 2021).
>
>
> **Q5: “There is no mention of code release by the authors.”**
>
> **A5**: Thank you for letting us know! We will have an official GitHub repo upon the acceptance of the paper. You may find our current codebase under the following anonymous link: https://drive.google.com/drive/folders/1-sm8lB6xQqDPEPowzoktLaikg7EVEDVJ?usp=sharing

---

### Official Review · Reviewer_KEao · 2022-10-31

**Confidence:** 3
**Correctness:** 3
**Technical Novelty And Significance:** 3
**Empirical Novelty And Significance:** 3
**Recommendation:** 6

**Clarity, Quality, Novelty And Reproducibility:**

Clarity: the paper was generally quite clear and well-written.  I have some minor comments here, like in Figure 1, the "snowflake" indicates parameters being frozen, but this wasn't explicitly mentioned anywhere; or in the setting about baselines, it's not explicitly mentioned which ones require finetuning and which ones don't.  But these are very minor and could easily be addressed between acceptance and final camera-ready version.

Quality and Reproducibility: No issues, these are both fine.

Novelty: This is related to my comment above regarding the weaknesses.  I find the observation of forward transfer in prompts to be novel.  The observation that catastrophic forgetting is not an issue with prompt engineering (as opposed to finetuning) seems not novel to me, as it's not really a contribution of this paper but more a contribution of the entire prompt engineering body of work prior to this paper.

**Strength And Weaknesses:**

Strengths:
- The paper is well written and generally easy to follow.
- The observation that parameterization of the prompt matters so much for performance is one that I had not appreciated before.
- The novel observation in this paper, that Progressive Prompting enables forward transfer is interesting.

Weaknesses:
For me, the main weakness of this paper---which prevents me from recommending acceptance---is that the connection to catastrophic forgetting (and thus, the motivation for many of the baselines reported in the paper) does not make sense.  Let me state this objection/confusion as clearly as possible so the authors may let me know if I have misunderstood, in which case I would be happy to amend my score.  As I see it, the problem of catastrophic forgetting in language models is a problem that only occurs in a fine-tuning setting.  In this work, the authors have specified the (common in continual-learning) setup in which the model knows the identity of the task both during training and during inference.  Since prompt tuning (either soft prompt-tuning or manual natural-language prompt tuning) has  now become standard practice for adapting pre-trained language models to a particular task, the standard thing to do for adapting a pretrained language model for multiple tasks is to learn a separate prompt for each of the tasks, and at inference time (since the task identity is assumed to be known) simply select the prompt corresponding to the given task and run the model using that prompt.  In such a setting, there is no forgetting since the model parameters themselves do not change.  Thus in some sense it seems like this paper is attempting to solve a problem which already does not exist in the prompt-engineering paradigm.  The baselines which involve retraining a full model (e.g. the baselines "finetune", "EWC", "A-GEM", "Experience replay", "MBPA++", "IDBR") don't seem relevant to me.

Having said the above, the "standard thing" that I mentioned is one of the benchmarks that the authors consider (Per-task prompts).  In my mind, the fact that the ProgressivePompts beats Per-task prompts is *the* interesting contribution of this paper, and (as the authors say) indicates some forward transfer between prompts.  This could be useful in many scenarios, especially in the low-data regime for subsequent tasks.  If the paper had been written with this as the main selling point (with all of the natural ablations and sanity-checks one would want), I would likely have been much more positive on it.

I think there are two ways in which I would be convinced to change my score:
1. The authors explain why my above argument is wrong, i.e. I have some misunderstanding and the problem of catastrophic forgetting is actually relevant in the prompt-engineering world; or,
2. The authors reframe their paper as being about the forward transfer phenomenon, and conduct more thorough experiments to explore that phenomenon (e.g., how does the transfer vary with dataset size for the subsequent tasks, how does it compare to a single prompt of length 2M (where M is the length of each individual prompt), etc.).


**Summary Of The Paper:**

This paper concerns language models in the pretrain-then-prompt-tune paradigm, and aims to introduce a prompting method to allow for  solving multiple tasks.  This can be described a continual learning setup, where the task ID is known to the model at both training and inference time.  The method, called Progressive Prompts, concatenates all prompts learned for previous tasks and learns a new task-specific prompt for each new task; thus the full prompt for task $N$ is the concatenation of the (previously-learned) prompts for tasks $1$ through $N-1$, concatenated with the newly-learned prompt for task $N$.

Evaluating their method on text classification tasks, the authors find favorable performance compared to their chosen set of baselines, which include both methods that fine-tune the full language model (with or without data replay/buffer) and other purely prompt-based approaches.

An additional finding of the paper is that parameterization of the prompt matters for performance; specifically, the authors parameterize the  (soft) prompt via a residual MLP and find via ablations that this parameterization improves performance.

**Summary Of The Review:**

I do not currently recommend the paper for publishing because its observations are presented as a solution to catastrophic forgetting, which does not seem to be a problem in the prompt-engineering paradigm.

---

Update post rebuttal: The authors have sufficiently addressed my concerns, and I have updated my score to an accept.

---

> ### Author Response · Authors · 2022-11-19
> **Response to Reviewer KEao**
>
> Thank you for your detailed review of our paper and constructive feedback! We have revised our paper and addressed your concerns. We summarize below the changes we made.
>
> **1. Emphasizing forward transfer**
>
> We agree that forward transfer is a very important component of Progressive Prompts on which we haven’t focused enough in our previous version.
>
> Firstly, we changed the title and reframed the paper to emphasize forward transfer. Secondly, we ran a comprehensive set of experiments based on your suggestions and dedicated Section 5.3 for it. We summarize below our new experiments.
>
> **“how does the transfer compare to a single prompt of length 2M (where M is the length of each individual prompt); how does the transfer vary with dataset size for the subsequent tasks”.**
>
> We have conducted new experiments on forward transfer to explore *whether learning a single prompt of size 2M is better than learning two progressive prompts of size M* on related tasks. We compare the performance of a single prompt (100 tokens) and Progressive Prompts trained on two tasks (50 tokens per task). We used six pairs of transfer learning tasks from similar domains (IMDb ⇔ SST2, Amazon ⇔ Yelp, QQP ⇔ MRPC). We perform experiments in full-shot (using all samples) and few-shot (2, 5 and 20 samples per class) settings with T5-large model. Summary of our results is shown below:
>
> ***Forward transfer results with 5 samples / class:***
> | Source task → Target task | Prompt Tuning (w/o transfer) | Progressive Prompt (w/ transfer) | Relative Improvement |
> |---------------------------|------------------------------|----------------------------------|----------------------|
> | IMDB → SST2               | 58.2 ± 2.7                   | **84.4 ± 8.1**                   | +45.0%               |
> | SST2 → IMDB               | 61.1 ± 5.0                   | **91.2 ± 2.5**                   | +49.3%               |
> | Amazon → Yelp             | 26.2 ± 3.8                   | **27.3 ± 1.4**                   | +4.2%                |
> | Yelp → Amazon             | 24.2 ± 2.1                   | **29.7 ± 3.7**                   | +22.7%               |
> | MRPC → QQP                | 88.4 ± 2.5                   | **92.0 ± 0.3**                   | +4.1%                |
> | QQP → MRPC                | **86.9 ± 0.6**               | 84.3 ± 0.6                       | -3.0%                |
> | Average                   | 57.5                         | **68.2**                         | +20.4%               |
>
> ***Average performance with/without transfer across different data limits:***
> |                      | 2/class  | 5/class  | 20/class | Full dataset |
> |----------------------|----------|----------|----------|--------------|
> | Prompt Tuning  (w/o transfer)       | 55.1     | 57.5     | 73.3     | 80.3         |
> | Progressive Prompt (w/ transfer)   | **60.8** | **68.2** | **77.8** | **81.2**     |
> | Relative Improvement | +12.9%   | +20.4%   | +9.8%    | +1.3%        |
>
> Full results / illustration are available in Section 5.3 of the latest revision, we also provide a screenshot below: https://drive.google.com/file/d/1w7F0UHDi1eejPe0DVO_d4Vi9lctGR4PY/view?usp=share_link
>
> We observe *significant forward transfer in low-data regime*: +20.4% and +12.9% average relative improvement with Progressive Prompts compared to prompt tuning in 5-shot and 2-shot settings respectively. In a full-shot setting, we observe modest improvement (+1.3% average relative improvement). Overall, we can see *forward transfer between Progressive Prompts in all data regimes*, and it is especially pronounced in the low-data setup.
>
> **2. The importance of catastrophic forgetting**
>
> **“As I see it, the problem of catastrophic forgetting in language models is a problem that only occurs in a fine-tuning setting.”**
>
> We would like to clarify that catastrophic forgetting is not limited to the fine-tuning setting, but also happens in prompt tuning-based continual learning. The current SOTA approach for continual learning in T5 (Qin et al, ICLR 2022), *continually trains a common prompt* shared across all tasks, and while it achieves forward transfer it also *suffers from forgetting*. Our approach, on the other hand, both achieves forward transfer and avoids forgetting. Our method establishes a new SOTA on standard benchmarks for continual learning in language models, which we summarize in Section 5.1/5.2.
>
> As to ***minor comments***: thank you for letting us know! We fixed all the mentioned issues in our latest revision (explain snowflake in the illustration, show which methods require full model tuning).

---

> ### Author Response · Authors · 2022-11-22
> **Thank you for the update, Reviewer KEao!**
>
> Thank you very much for the update! We are glad that we have addressed your concerns, and believe that your comments helped to improve our paper a lot.

---

### Official Review · Reviewer_vg4X · 2022-11-02

**Confidence:** 3
**Correctness:** 4
**Technical Novelty And Significance:** 2
**Empirical Novelty And Significance:** 3
**Recommendation:** 8

**Clarity, Quality, Novelty And Reproducibility:**

Quality and clarity seem both good.

Novelty seems to fall under incremental from technical perspective -- the method to use expansion on model parameter for CL may be explored before by a few researchers.

**Strength And Weaknesses:**

Strength:
1) The writing seems very good that makes paper easy to follow.
2) The forgetting can be alleviated as the prompt for previous tasks are unchanged.
3) Transfer of knowledge from the previous task to new task is achieved by the new approach.
4) A lot better results on the experiments.

Question:
1) Should we compare the AdapterCL or other adapter model that could add a new part / parameters for each task, for the empirical results?
2) For the eval results in Section 5.1 and 5.2, the progressive prompt on T5 has much larger gap than the other base lines. The BERT model with progressive prompt has not as large enhancement. Are there some root cause? Let me know if my understanding are not good.

**Summary Of The Paper:**

The authors propose the progressive prompts that add a new prompt for each task and fix all other parameters from both feature extractor e.g. BERT, T5 and the prompt from previous tasks.

The new method should be able to both transfer knowledge from the previous task to new task, as well as reduce the forgetting as the prompt for previous tasks are unchanged.

**Summary Of The Review:**

The recommendation would be accept but marginally above the threshold. My reason for not a higher score results from the somewhat incremental on the technical novelty. My reason for not a lower level results from large gap on the performance compared to other method.

---

> ### Author Response · Authors · 2022-11-19
> **Response to Reviewer vg4X**
>
> Thank you for the review, comments, and constructive feedback! We are encouraged that you found our idea promising, our method effective and our writing clear. We provide answers to your comments and questions below.
>
> **Q1. “Should we compare the AdapterCL or other adapter model that could add a new part / parameters for each task, for the empirical results?”**
>
> **A1**: Thank you for the question! One of our baselines – LFPT5 (Qin et al., ICLR 2022) – reports better performance (+2.2% improvement on a sequence of classification tasks) than Adapter-Fusion (Pfeiffer et al., EACL 2021), which is a strong adapter model for continual learning tasks. Hence, we performed a comparison directly with LFPT5 as a superior approach over adapter methods. Our approach, Progressive Prompts, achieved +22.4% improvement over LFPT5 (in average accuracy after learning all tasks) on a standard continual learning benchmark for T5 model (Table 1). We updated Section 5.1 in our revised version to discuss adapter models.
>
> **Q2. “For the eval results in Section 5.1 and 5.2, the progressive prompt on T5 has much larger gap than the other baselines.“**
>
> **A2**: Section 5.1 focuses on few-shot learning experiments (standard setup for T5), while Section 5.2 investigates the performance on the full dataset (standard setup for BERT). In the few-shot setting, Progressive Prompt significantly outperforms the previous state-of-the-art method, hence we observe the larger gap.
>
> In Section 5.3, we perform both few-shot and full-shot experiments on a sequence of 15 tasks with BERT and T5 models. We observe that Progressive Prompt consistently achieves larger performance improvements under few-shot settings for both BERT and T5. Here is a summary of the results:
> |               |          T5         |     T5    |         BERT        |    BERT   |
> |---------------|:-------------------:|:---------:|:-------------------:|:---------:|
> |               | few-shot (20/class) | full-shot | few-shot (20/class) | full-shot |
> | Previous SOTA | 54.3                | 69.2      | 36.8                | 52.2      |
> | Prog Prompt   | **76.2**            | **79.5**  | **53.5**            | **69.3**  |
> | Delta         | 21.9                | 10.3      | 33.3                | 17.1      |
>
> We incorporated the explanation of these performance differences in our latest version of the paper (Section 5.1).
>
> **Q3: “Novelty seems to fall under incremental from technical perspective”**
>
> **A3**: We would like to clarify that we have two important contributions in this work:
> 1. We are the first to propose using prompt tuning in a progressive set-up and show that it allows forward transfer, while being totally resistant to forgetting.
> 2. We demonstrated the effectiveness of our method on many standard model architectures such as BERT and T5, and in long-sequence continual learning scenarios. Specifically, Progressive Prompt shows 21.9% and 33.3% improvements over the previous SOTA methods when learning on a sequence of 15 tasks in a few-shot setup.
>
> Hence, we believe that this paper proposes a simple yet very effective method that significantly outperforms previous state-of-the-art methods on various benchmarks and model architectures. We believe that there is great value in simplicity to address the challenges in continual learning.

---

### Decision · Program_Chairs · 2023-01-20

**Decision:**

Accept: poster

**Justification For Why Not Higher Score:**

The paper is the first step towards doing continual learning in LLMs and although interesting does not fully solve the problem or provide new insights in general for CL. The assumption of having access to task boundaries is a limitation.

**Justification For Why Not Lower Score:**

See strength section of the meta review

**Metareview: Summary, Strengths And Weaknesses:**

The paper focuses on continual learning in LLMs using prompting (in-context learning)

Strength:
- The writing seems very good that makes paper easy to follow.
- The forgetting can be alleviated as the prompt for previous tasks are unchanged.
- Transfer of knowledge from the previous task to new task is achieved by the new approach.
- strong empirical results
- The novel observation in this paper, that Progressive Prompting enables forward transfer is interesting.

Weakness:
-  The task-boundary assumption is a limitation of the paper
- The main focus of the paper is on the final average performance (e.g., accuracy, F1) of the models/algorithms. However, it is important to compare continual learning methods from different perspectives that measure the evolution of the accuracies and forward and backward transfers.



**Note From Pc:**

if the above contains the word "oral" or "spotlight" please see: "oral" presentation means -> notable-top-5% and "spotlight" means -> notable-top-25%. As stated in our emails, we are disassociating presentation type from AC recommendations

**Summary Of Ac-Reviewer Meeting:**

The paper was borderline and we set a meeting to discuss this. The main point that was discussed was the many assumptions in the paper and that the setting not being natural. However we focused on the work being a good first step in performing continual learning using in-context learning and therefore focusing on the positive side and acceptance.